# Bioinformatics analysis for the role of CALR in human cancers

Yijun Li[1], Xiaoxu Liu[1], Heyan Chen[1], Peiling Xie[1], Rulan Ma[2], Jianjun He[1]☯*, Huimin Zhang[1]☯*

1 Departments of Breast Surgery, The First Affiliated Hospital of Xi'an Jiaotong University, Xi'an, People's Republic of China, 2 Departments of Surgical Oncology, The First Affiliated Hospital of Xi'an Jiaotong University, Xi'an, People's Republic of China

☯ These authors contributed equally to this work.
* chinahjj@163.com (JH); huimin.zhang@xjtu.edu.cn (HZ)

**Data Availability Statement:** The datasets generated and/or analysed during the current study are available in the GeneCards repository [http://www.genecards.org], UALCAN repository[http://ualcan.path.uab.edu/analysis.html],GEPIA

## Abstract

Cancer is one of the most important public health problems in the world. The curative effect of traditional surgery, radiotherapy and chemotherapy is limited and has inevitable side effects. As a potential target for tumor therapy, few studies have comprehensively analyzed the role of CALR in cancers. Therefore, by using GeneCards, UALCAN, GEPIA, Kaplan-Meier Plotter, COSMIC, Regulome Explorer, String, GeneMANIA and TIMER databases, we collected and analyzed relevant data to conduct in-depth bioinformatics research on the CALR expression in Pan-cancer to assess the possibility of CALR as a potential therapeutic target and survival biomarker. We studied the CALR expression in normal human tissues and various tumors of different stages, and found that CALR expression was associated with relapse free survival (RFS). We verified the expression of CALR in breast cancer cell lines by vitro experiments. Mutations of CALR were widely present in tumors. CALR interacted with different genes and various proteins. In tumors, a variety of immune cells are closely related to CALR. In conclusion, CALR can be used as a biomarker for predicting prognosis and a potential target for tumor molecular and immunotherapy.

## Introduction

Cancer is one of the most important public health problems in the world. According to the Cancer statistics 2021 report [1], there were 19.3 million new cases and 10 million cancer deaths worldwide in 2020.The curative effect of traditional surgery, radiotherapy and chemotherapy is limited and has inevitable side effects. As a new type of therapy, molecular targeted therapy can interfere with specific molecules to prevent the growth, progression and metastasis of tumors. Compared with traditional chemotherapy, molecular targeted therapy has the advantage of being able to deliver drugs with high specificity and low toxicity. Consequently, it is of great significance to find ideal targets and new biomarkers for early diagnosis, improving prognosis and developing molecular targeted therapy for cancers.

Calreticulin (CALR) is a ubiquitous and highly conserved protein in cells. It was initially identified as a protein with the high affinity for calcium ions in the sarcoplasmic reticulum of

repository[http://gepia.cancer-pku.cn/index.html], Kaplan-Meier Plotter repository[http://kmplot.com/ analysis], COSMIC repository[https://cancer. sanger.ac.uk/cosmic/], Regulome Explorer repository[http://explorer.cancerregulome.org/], STRING repository[https://string-db.org/], GeneMANIA repository[(http://www.genemania. org], TIMER repository[https://cistrome.shinyapps. io/timer/].

**Funding:** Huimin Zhang was supported by the National Natural Science Foundation of China (No. 81702632).The funders had no role in study design, data collection and analysis, decision to publish, or preparation of the manuscript.

**Competing interests:** The authors have declared that no competing interests exist.

**Abbreviations:** ACC, Adrenocortical carcinoma; APC, antigen-presenting cells; BC, breast cancer; BRCA, breast invasive carcinoma; BLCA, bladder urothelial carcinoma; CALR, calreticulin; CNS, CNS; CESC, cervical squamous cell carcinoma; CHOL, cholangiocarcinoma; COAD, colon adenocarcinoma; CUC, cervical cancer; DC, dendritic cells; DLBC, Lymphoid Neoplasm Diffuse Large B-cell Lymphoma; ESCA, esophageal carcinoma; GBM, glioblastoma multiforme; HNSC, head and neck squamous cell carcinoma; ICD, immune cell death; KICH, kidney chromophobe; KIRC, kidney renal clear cell carcinoma; KIRP, kidney renal papillary cell carcinoma; LIHC, liver hepatocellular carcinoma; LUAD, lung adenocarcinoma; LUSC, lung squamous cell carcinoma; OV, Ovarian serous cystadenocarcinoma; PAAD, pancreatic adenocarcinoma; PCPG, pheochromocytoma and paraganglioma; PRAD, prostate adenocarcinoma; READ, rectum adenocarcinoma; RFS, relapse free survival; SARC, sarcoma; SKCM, skin cutaneous melanoma; STAD, stomach adenocarcinoma; TGCT, testicular germ cell tumors; THCA, thyroid carcinoma; THYM, thymoma; UCEC, uterine corpus endometrial carcinoma.

skeletal muscle [2–4]. Recent studies have shown that CALR is involved in the occurrence, proliferation [5, 6], migration and adhesion [7, 8] of the tumor, and mediates cell phagocytosis [9], signal transduction and immune cell death (ICD) [10]. By activating dendritic cells (DC) and cytotoxic T cells, CALR leads to tumor cell ICD [11, 12] as phagocytic signals. It has been reported that compared with normal tissues, the expression level of CALR in colorectal cancer [13] vaginal cancer [14], oral cancer [5], breast ductal carcinoma [15, 16] and PRAD [17] was increased. Moreover, overexpression of CALR has been considered as a reliable biomarker for detecting urothelial carcinoma and predicting the prognosis [18].

As a potential target for tumor therapy, the expression of CALR in most tumors remains unclear and few studies have comprehensively analyzed the role of CALR in cancers. Therefore, according to several large public databases, we conducted in-depth bioinformatics research on the CALR expression in Pan-cancer to assess the possibility of CALR as a potential therapeutic target and survival biomarker, which provided additional information for exploring the mechanism of tumor progression, predicting the prognosis and researching new targeted drugs.

# Materials and methods

## GeneCards

GeneCards (www.genecards.org) summarizes human gene annotation data comprehensively and authoritatively. Based on genes, it automatically mines and integrates from more than 80 digital sources to form a web-based deep link card that can be used for more than 73,000 human gene entries [19]. In our study, we obtained the CALR mRNA expression information in normal human tissues in GTEx, BioGPS, and SAGE databases according to the GeneCards website.

## UALCAN

UALCAN (http://ualcan.path.uab.edu/analysis.html) is an open, interactive network resource that provides data analysis of The Cancer Genome Atlas (TCGA) and MET500 cohort [20]. In our study, we compared the differential expression of CALR in human normal tissues and cancers by UALCAN database analysis. The p-value was generated by the Student's t-test and p-value < 0.05 was considered statistically significant.

## RT-qPCR

Total RNA was extracted, cDNA was obtained using the RNA Fast 200 (Fastagen Biotech; 220010), and converted to cDNA with the PrimeScript™ RT Master Mix (Takara Biotechnology; RR036A). RT-qPCR analysis was performed using PCR primers with the following sequences: CALR, 5′– TCG ACA ACC CAG ATT ACA AGG –3′ and 5′– AAG ATG GTG CCA GAC TTG AC –3′; and 18S rRNA, 5′– GGA CAG GAT TGA CAG ATT GAT AGC –3′ and 5′– TGC CAG AGT CTC GTT CGT TA –3′, with the SYBR Premix Ex Taq™ II (Takara Biotechnology; RR820A) in Bio-Rad CFX96 system (Hercules).

**Immunoblot assay.** Whole-cell lysates were prepared in RIPA lysis buffer containing a mixture of protease inhibitors. Proteins were separated using SDS-PAGE, blotted onto polyvinylidene difluoride membranes (Millipore), and probed with primary antibodies against CALR (Abcam) and GADPH (Proteintech) at 4˚C overnight. HRP-conjugated secondary antibody (Proteintech) was used. Signals were detected using electrochemiluminescence (Bio-Rad) by the chemiluminescence reagent (Millipore).

## GEPIA

GEPIA (http://gepia.cancer-pku.cn/index.html) is an intuitive network application tool from the information of the TCGA and GTEX databases, 9,736 tumors tissues and 8,587 normal samples, and using the standard processing of RNA sequencing data for the output of gene expression analysis [21]. We used GEPIA to explore the expression of CALR in different pathological stages of tumors. P < 0.05 was considered to be the significant different.

## Kaplan-Meier Plotter

The Kaplan-Meier Plotter (http://www.kmplot.com/analysis/) is an online tool for drawing survival curves based on GEO, EGA and TCGA databases [22]. By calculating the 95% confidence interval and P-value, it compared the difference of survival rate between the high-expressed group and low- expressed group. We analyzed the effect of CALR expression on RFS (relapse-free survival) in different cancers by this website.

## COSMIC

COSMIC (https://cancer.sanger.ac.uk/cosmic/) is a high-resolution resource for studying the targets and trends of human cancer genetics. Combining the whole genome sequencing results of tumors with individual publications, it showed the information of coding mutations, non-coding mutations, gene fusions, genome rearrangements, abnormal copy number segments, abnormal expression variants and differentially methylated CpG dinucleotides. At present, COSMIC is the most extensive database of cancer mutations in the world [23]. In this study, we used the COSMIC database to analyze the mutation types of CALR in different types of cancers.

## Regulome Explorer

Regulome Explorer online tools (http://explorer.cancerregulome.org/) can visually evaluate CALR expression in cancers and its correlation with other genes in the TCGA database. The pairwise correlation of the two genes was calculated by Spearman's correlation analysis and displayed by the circus diagram. In our study, we only showed genes with p-value > log10.

## STRING

STRING (https://string-db.org/) database is an extensive, objective global network which is designed to ingather, integrate and score the published protein-protein interaction (PPI) information, and to supplement this data by scientific calculations and predictions [24]. We built a PPI network associated with CALR by the STRING database.

## GeneMANIA

GeneMANIA (http://www.genemania.org) is a user-friendly and flexible website for proposing gene function hypotheses, determining gene priority for the functional analysis, and generating analysis gene list [25]. It can find and predict proteins with similar functions based on a large number of genomics and proteomics data. We explored genes that may interact with CALR according to the GeneMANIA website.

## TIMER

TIMER (https://cistrome.shinyapps.io/timer/) is an integrated computing tool which is aimed at researching and visualizing genomics data and tumor immunology [26]. The TIMER

algorithm could explore the relationship between target gene expression in tumor cells and immune infiltration according to the Spearman's test, and draw the scatter diagram. Spearman's Rho value and statistical significance are displayed in the upper left corner of the diagram. All p values were two-sided and a p value of $<0.05$ was considered statistically significant. In this study, TIMER was used for analyzing the relationship between CALR and the immune cell infiltration in different tumors.

In our study, firstly we introduced the molecular characteristics and differential expression of CALR in tumors and normal tissues. The survival analysis was performed between the expression level of CALR and clinical prognosis. The above studies were used to confirm the role of CALR in the occurrence of a variety of tumors. Subsequently, we performed mutation, Genome-wide association and PPI analysis to explore the possible molecular mechanism of CALR carcinogenesis. Finally, we analyzed the relationship between CALR and immune cells in different tumor cells to explore the CALR related tumor immune mechanism and the possibility of CALR as a target of immunotherapy.

The Ethics Committee of the First Affiliated Hospital of Xian Jiaotong University exempted the review of the study because all these databases are publicly available.

## Results

### CALR mRNA in normal tissues

In order to explore the expression pattern of CALR under physiological conditions, we detected the expression of CALR mRNA in all types of normal tissues, which can provide clues for the function of the gene. The expression of CALR mRNA in human normal tissues was analyzed according to GTEx, BioGPS, and SAGE databases from the GeneCards website. As shown in Fig 1, CALR mRNA was expressed differently in different tissues, with the highest expression in the liver according to the GTEx database and in smooth muscle based on the BioGPS database. While in the SAGE databases, the kidney is the tissue with the highest CALR mRNA expression in the human body.

### The expression of CALR in cancers

To investigate the role of CALR in cancers, we detected the CALR expression in 23 types of tumors and corresponding normal tissues using the UALCAN database. As shown in Fig 2A and Table 1, CALR was up-regulated in 16 types of tumors compared with the corresponding normal tissues, which indicated that CALR may play an oncogenic role in most cancer types. However, in thyroid carcinoma (THCA), the expression of CALR decreased and there was no significant change of expression in the other five tumors, which may be related to different carcinogenic mechanisms in different tumors.

Since breast cancer(BC) is the most common malignancy worldwide [1], we further tested the mRNA and protein expression of CALR in immortalized breast epithelial cells (MCF-10A) and six different human BC cell lines, which encompassed the major clinical categories of BC based on expression of the estrogen receptor (ER), progesterone receptor (PR) and human epidermal growth factor receptor 2 (HER2): MCF-7 is ER+PR+ HER2− (Luminal A molecular subtype); HCC1954 and MDA-MB-453 are ER−PR−HER2+ (HER2-enriched subtype), and MDA-MB-231, BT549 and SUM159 are ER−PR−HER2− (Basal-like subtype) [27]. Reverse transcription (RT) and quantitative real-time PCR (qPCR) and immunoblot (IB) assays revealed that CALR mRNA and protein were more highly expressed in all 6 BC cell lines compared to MCF-10A cells, and highest expression was observed in the Basal-like cell lines (Fig 2B).

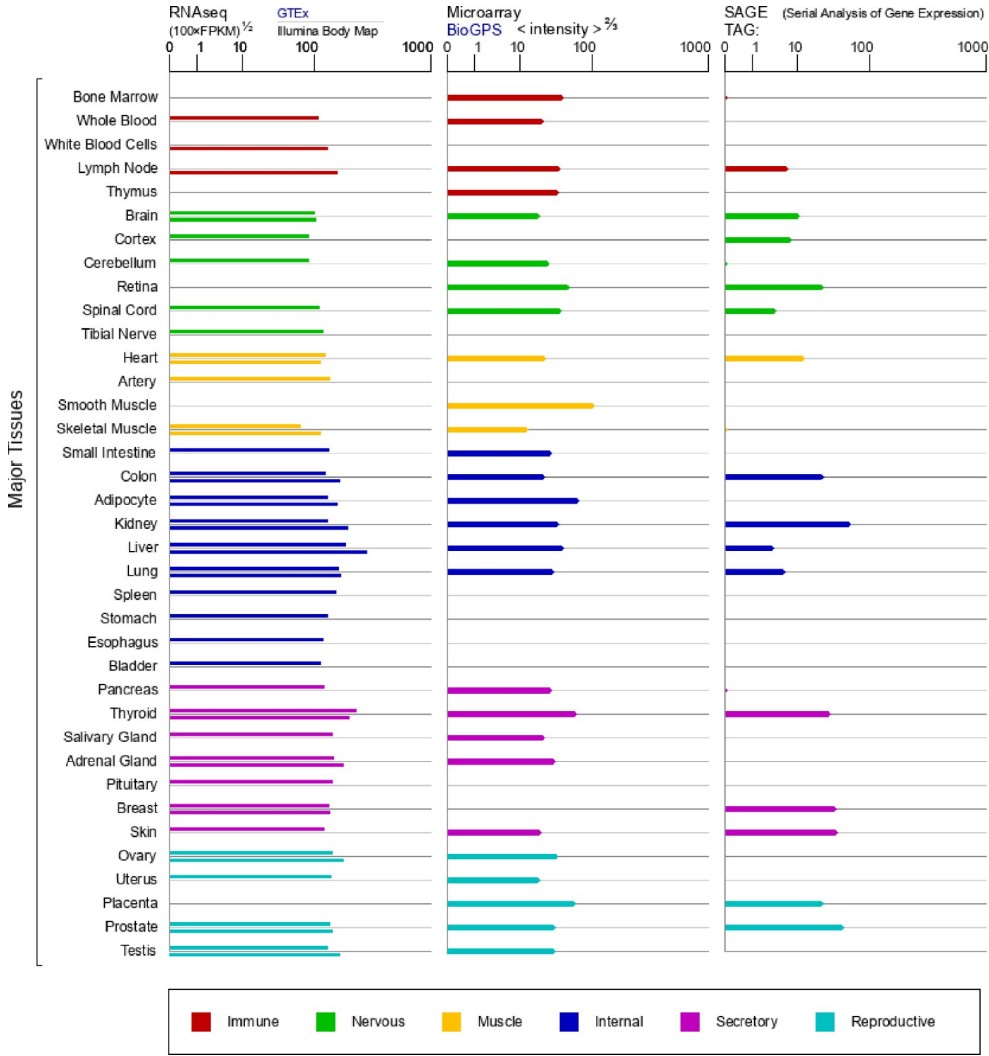

**Fig 1. The CALR mRNA expression in normal human tissues in GTEx, Illumina, BioGPS, and SAGE databases.**

## Correlation of CALR expression and pathological stages of tumors

In order to determine whether CALR is associated with tumor progression, we further evaluated the relationship between the CALR and tumor pathological stages based on the GEPIA. The expression of CALR increases with tumor progression in bladder urothelial carcinoma (BLCA), breast invasive carcinoma (BRCA) and kidney renal clear cell carcinoma (KIRC). While in colon adenocarcinoma (COAD) and THCA, down-regulated CALR is associated with the higher tumor stage (Fig 3). This indicates different correlation between CALR and tumor progression in different tumors. In BLCA, BRCA and KIRC, higher expression of CALR was associated with more advanced tumor which could be used as an indication of rapid tumor progression.

## CALR expression and RFS of cancers

To further investigate the relationship between CALR expression and tumor prognosis, we used the Kaplan-Meier Plotter to study the correlation between CALR and relapse-free survival (RFS). According to the results of Kaplan-Meier analysis, higher CALR expression correlated

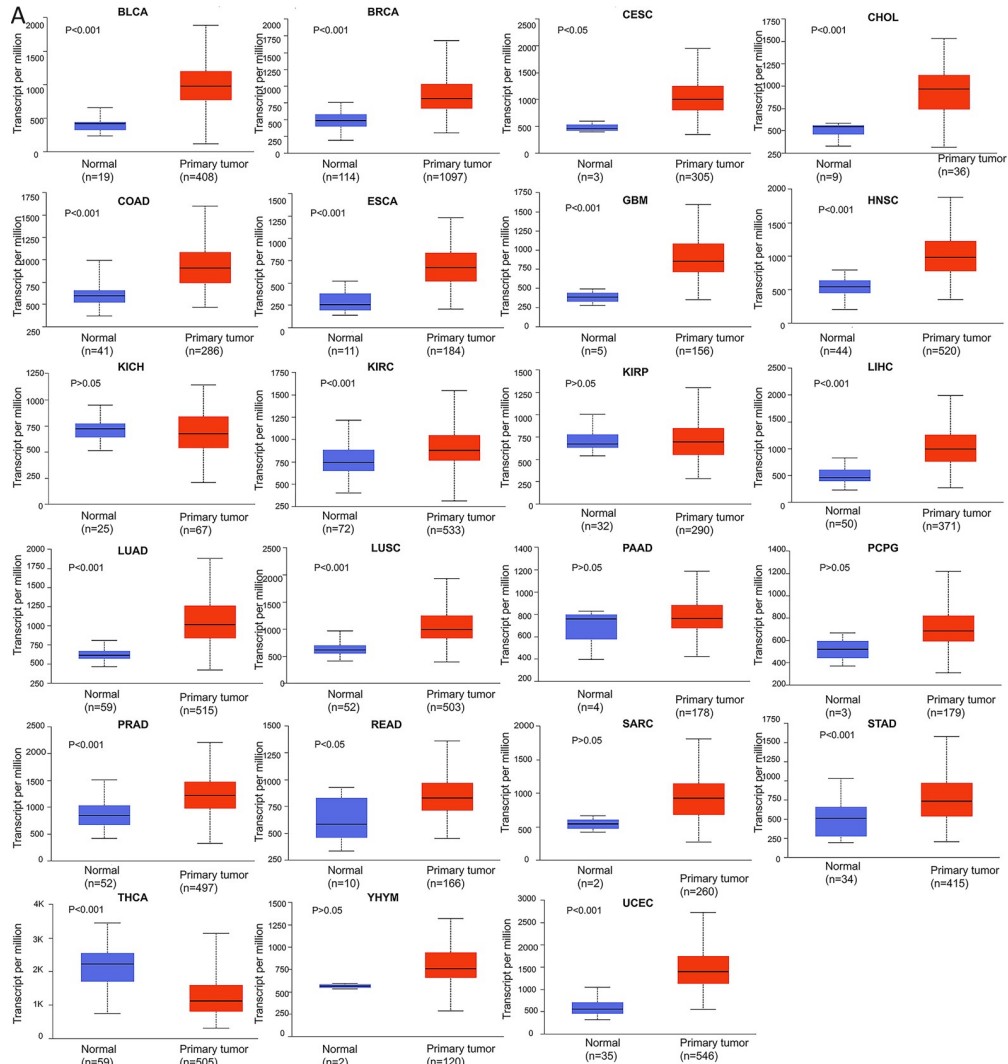

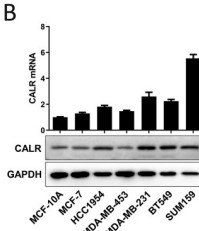

**Fig 2. The differential expression of CALR in human normal tissues and cancers by UALCAN database analysis.**
Note: P value< 0.05 is considered statistically significant.

with worse RFS in KIRC, liver hepatocellular carcinoma (LIHC), lung squamous cell carcinoma (LUSC), kidney renal papillary cell carcinoma (KIRP) and sarcoma (SARC). However, in ovarian serous cystadenocarcinoma (OV) and THCA, higher expression of CALR tends to have better RFS (Fig 4). The above data suggests that CALR expression exhibit different effect on patients' survival among tumors, which indicates a different role played by CALR in the biological characteristics among different tumors.

**Table 1. The expression level of CALR in tumors analysis compared with normal tissue by UALCAN database.**

| Expression level compared with normal tissue | Tumor type |
|---|---|
| **Higher** | BLCA, BRCA, CESC, CHOL, COAD, ESCA, GBM, HNSC |
| | KIRC, LIHC, LUAD, LUSC, PRAD, READ, STAD, UCEC |
| **Lower** | THCA |
| **No statistically significant** | KICH, KIRP, PAAD, PCPG, SARC, THYM |

Note: P value< 0.05 is considered statistically significant.

## CALR mutations in cancers

The study of gene mutation in tumors can provide a possible direction for etiology research and targeted therapy. Therefore, we performed mutation type analysis in various types of tumors,

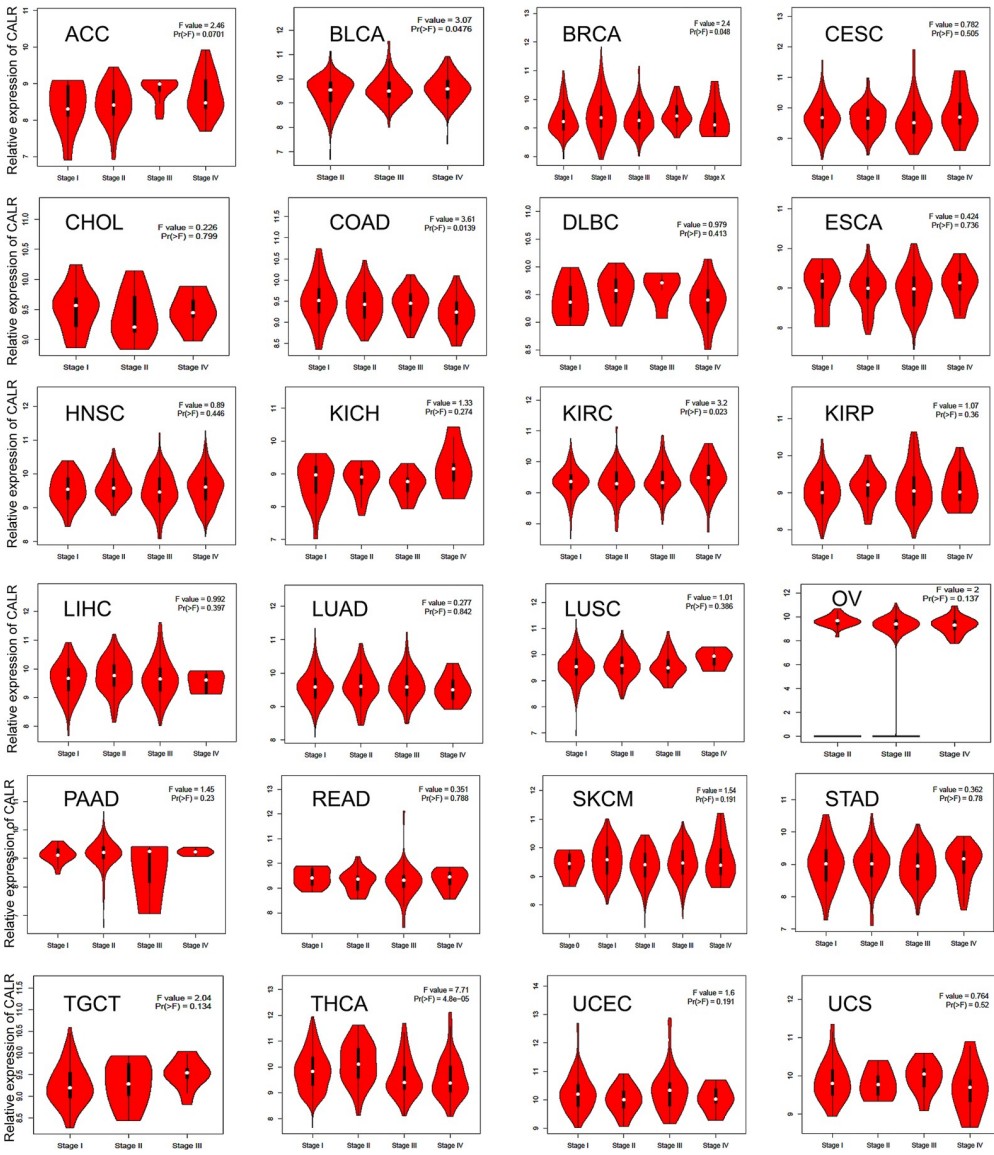

**Fig 3. Correlation between CALR expression with pathological stages of tumors (GEPIA).** Note: P value< 0.05 is considered statistically significant.

including nonsense, missense, synonymous and complex mutations, frameshift insertion, inframe and frameshift deletion and others (Fig 5 and Table 2). The missense mutation occurred most frequently and was detected in 17 types of tumors. Among the base-pair mutations, G > T, C > T and G > A mutations were most common, which were observed in 13, 10 and 10 cancer types, respectively (S1 Fig). According to cBioPortal, we detected 34 mutations between 0 and 417 amino acids (S2A Fig). In the TCGA database, ovarian cancer, uterine cancer, uterine carcinosarcoma and mesothelioma had higher mutation levels (S2B Fig). These results suggest that most tumors with high expression of CALR have a high tumor mutation burden.

## Genome-wide association of CALR in cancers

In order to study the molecular mechanism and function of the CALR gene in tumorigenesis, we conducted the genome-wide association analysis on CALR and drew the circus diagram to display the interaction network between CALR and other genes in different tumors (Fig 6). In the circus diagram, circular layout edges show the relevance, the outer loop shows cytogenetic bands and the inner loop indicates associations with features of lacking genomic coordinates. The connected curve represents a pair of genes with P value <- log10 of the correlation between DNA methylation, somatic copy number, somatic mutation and protein level according to Spearman's correlation analysis. As shown in Fig 6, a large number of genes are significantly correlated with CALR detected in BRCA, THCA, esophageal carcinoma (ESCA), stomach adenocarcinoma (STAD), KIRC, and glioblastoma multiforme (GBM), indicating that CALR is closely related to the other genes in the genome of these types of cancers. The specific genes associated with CALR in different tumors and their correlations are shown in S1 Table.

## PPI network of CALR

The PPI network can provide a basis for exploring the biological behavior of CALR in carcinogenesis. According to the analysis of binding protein with CALR screened by String tool, we obtained a total of 21 corresponding proteins supported by experimental evidence (Fig 7A). Subsequently, we studied the interactive network of CALR with other genes through GeneMANIA database (Fig 7B). Through the cross analysis of the above two databases, we found 5 common members—PDIA3, B2M, GANAB, CANX, TAPBP, which suggests the biological behavior of CALR may be related to the above 5 genes/proteins.

## Immune cell infiltration of CALR in cancers

Previous studies have shown that CALR can participate in several aspects of tumor immune regulation [10–12, 28]. Thus, we used the Timer database to study the correlation between CALR and 6 kinds of immune cell infiltrations in 39 types of tumors, including B cells, CD8 [+] T cells, CD4 [+] T cells, macrophages, neutrophils and DCs (S3 Fig). According to the Cancer statistics 2020 report [29], the highest incidence rate of cancer for females is breast cancer, lung cancer and colorectal cancer. For males, the 3 most common cancers are prostate, lung and bronchus, and colorectal. Therefore, we list the results of immune cell infiltration in these tumors in Fig 8. Among the five most common human tumors, CALR is correlated with 4 kinds of immune cells in breast cancer and colon cancer (p<0.05). These findings suggest that the carcinogenic mechanism of CALR may be related to anti-tumor immunity, and CALR may have potential to impact the immunotherapy.

## Discussion

As a multifunctional protein, CALR was previously considered to be a resident protein of the sarcoplasmic reticulum, with the characteristics of $Ca^{2+}$ buffer and molecular chaperone. Since

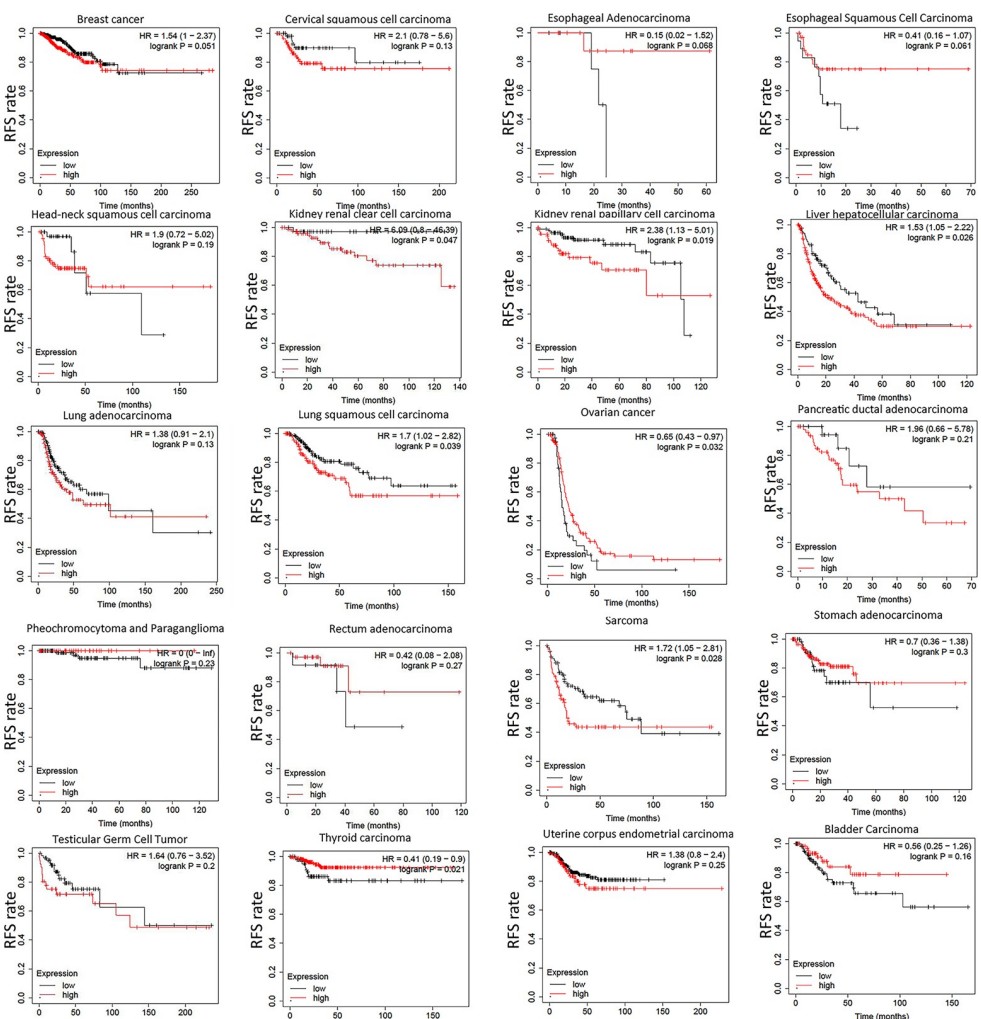

**Fig 4. Kaplan-Meier analysis of the association of CALR expression with RFS in different cancers.** Note: P value< 0.05 is considered statistically significant. RFS, relapse-free survival.

scientists found that the conditional medium containing CALR released from cultured cells can kill tumor cells and reduce angiogenesis [30, 31], and some studies suggested that CALR may participate in the clearance of tumor cells by activating the immune system [32, 33], the role of CALR played in tumor progression and anti-tumor immunity gained more and more attention. However, the effect of CALR on tumor prognosis, its relationship with the immune system and the underlying mechanisms in pan-cancer has not been well characterized. Therefore, we applied bioinformatics analysis to verify the possibility of CALR as an emerging and accurate tumor biomarker for targeted therapy.

As shown in Fig 1, the distribution of CALR in normal tissues is different, which may be related to the presence of CALR in the sarcoplasmic reticulum. Therefore, CALR is highly expressed in ER-rich tissues or tissues involved in a large number of protein syntheses, such as the liver, kidney, and thyroid.

CALR is involved in many important aspects of cancer, including cell proliferation, adhesion and migration, phagocytosis, integrin signal transduction, ICD, etc. The increase of CALR level is highly correlated with the occurrence of different types of cancer. For example,

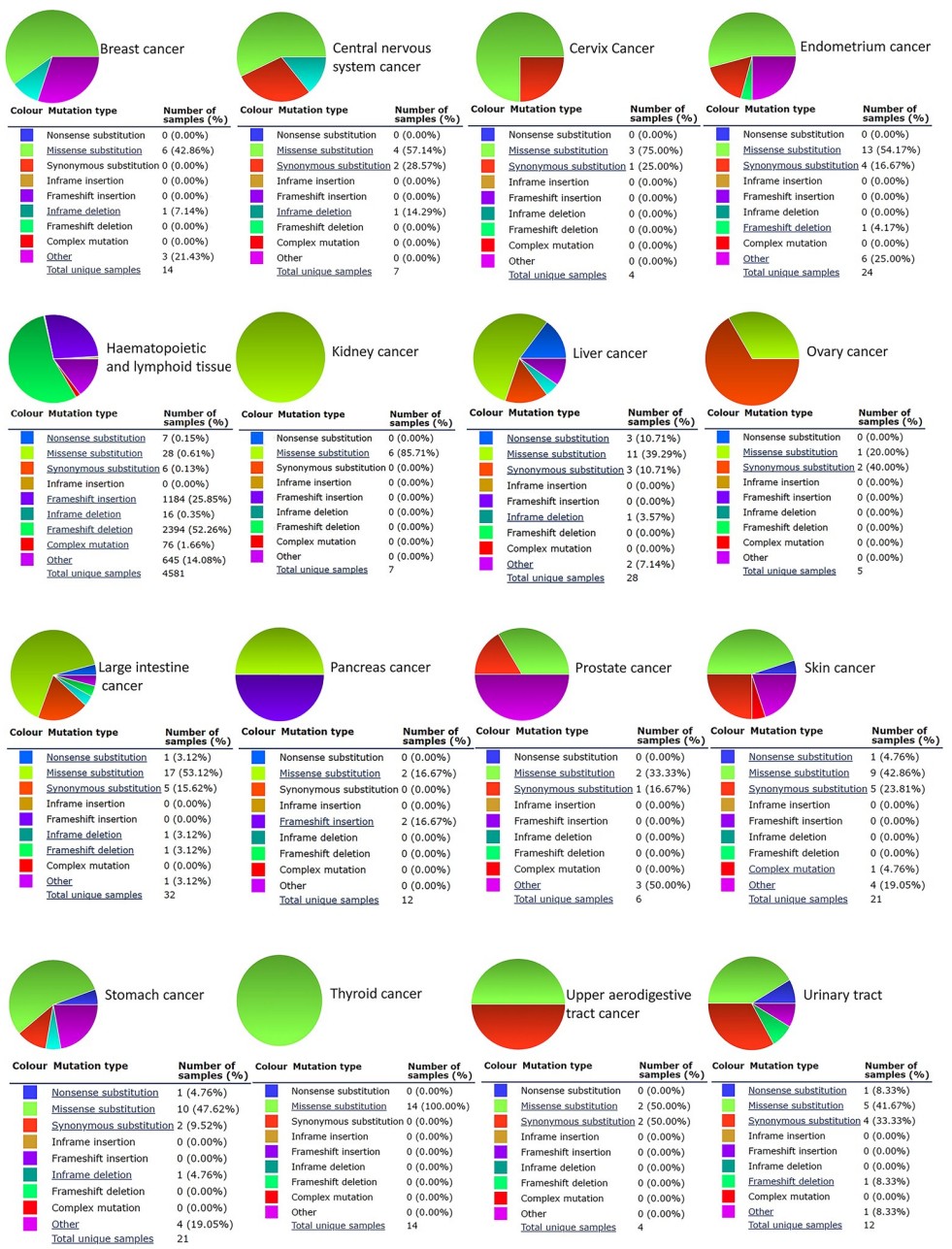

**Fig 5. Pie chart showing the percentage of different CALR mutation types in tumors (COSMIC).**

overexpression of CALR has been found in breast cancer [16], bladder cancer [34], PRAD [17], gastric cancer [35], hepatocellular carcinoma [36], colon cancer [37], pancreatic cancer [38], melanoma [39], esophageal cancer [40] and leukemia [41]. Moreover, the increase of CALR is more likely to cause tumor metastasis, and the mechanism can be explained by that as a major calcium homeostasis regulator, CALR participates in tumor metastasis through regulating $Ca^{2+}$ signal to induce cell migration [42]. Another possible reason is that CALR actively regulates cell migration and cell survival in the anoikis which is caused by the lack of matrix attachment [43]. In addition, higher expression of CALR is also associated with the more

**Table 2. CALR mutation types in tumors (COSMIC).**

| Mutation types | Tumor type |
|---|---|
| **Nonsense substitution** | HNSC, COAD, LIHC, STAD, BLCA, LUAD, SKCM |
| **Missense substitution** | CNS cancer, HNSC, KICH and KIRC, LUAD, PCPG, COAD, SKCM |
| | LIHC, OV, PRAD, STAD, BRCA, CUC, UCEC, BLCA, THCA, ESCA |
| **Synonymous substitution** | CNS cancer, UCEC, LUAD, COAD, SKCM, BLCA, LIHC, OV |
| | STAD, ESCA, CUC, HNSC, PRAD |
| **Frameshift insertion** | HNSC, PCPG |
| **Inframe deletion** | HNSC, CNS cancer, STAD, COAD, LIHC, BRCA |
| **Frameshift deletion** | UCEC, HNSC, COAD, BLCA |
| **Complex mutation** | SKCM, HNSC |
| **Other** | COAD, PRAD, SKCM, STAD, BLCA, BRCA, UCEC, HNSC, LIHC |

aggressive malignant processes and poorer prognosis in esophageal cancer [37], gastric cancer [7] and breast ductal carcinoma [44], which is also confirmed in our study. In most tumors,

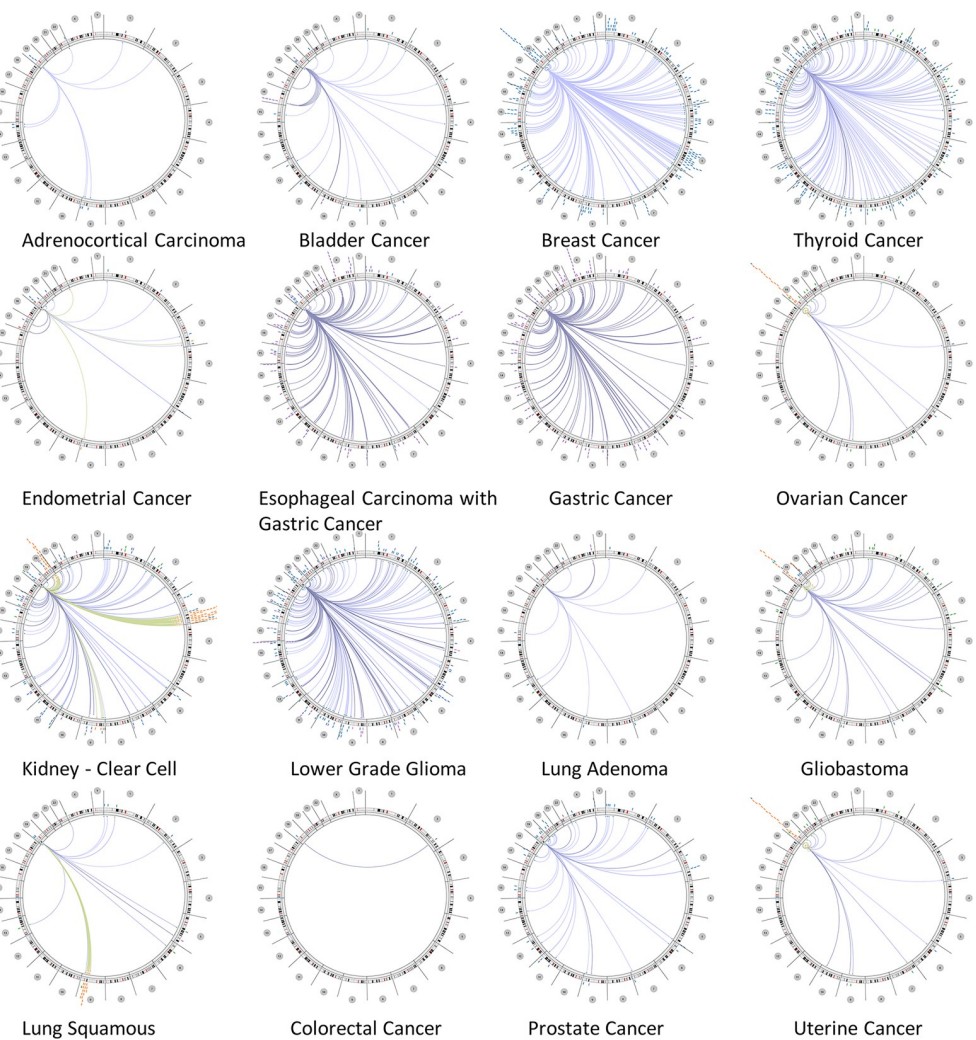

**Fig 6. Circus diagram showing the correlation between CALR and other genes from the TCGA database.**

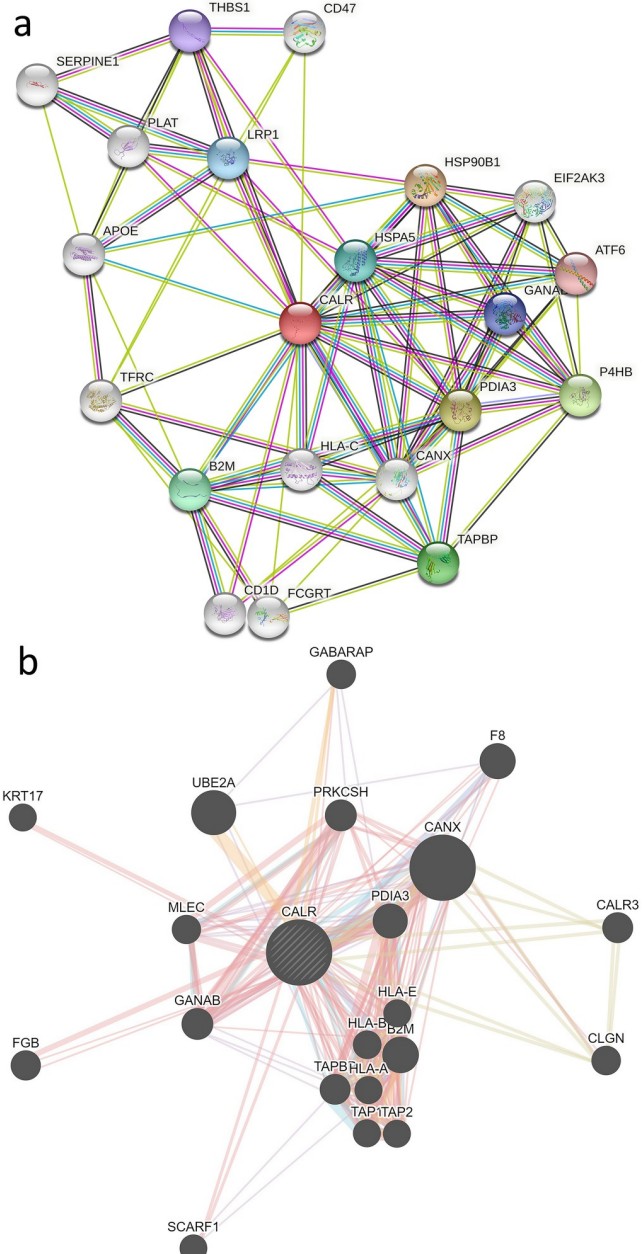

**Fig 7. Protein-protein interaction network of CALR.** (a)The top 21 proteins associated with CALR based on the STRING database. (b)The interacted genes with CALR according to the GeneMANIA website.

overexpression of CALR was associated with the higher stage and worse prognosis. Notably, the opposite situation existed in individual tumors, which may be caused by different mutation types (Fig 5) and associated genes (Fig 6) of CALR in different cancer types and needs further validation experimentally.

In order to study the molecular mechanism of CALR involved in tumor progression, we used String and GeneMANIA databases finding that PDIA3, B2M, GANAB, CANX, and TAPBP interact with CALR. It has been found that PDIA3 and CALR are highly co-expressed in micrometastasis pancreatic cancer cell lines PC-1 and Capan-2 [45]. DNMT inhibitors can

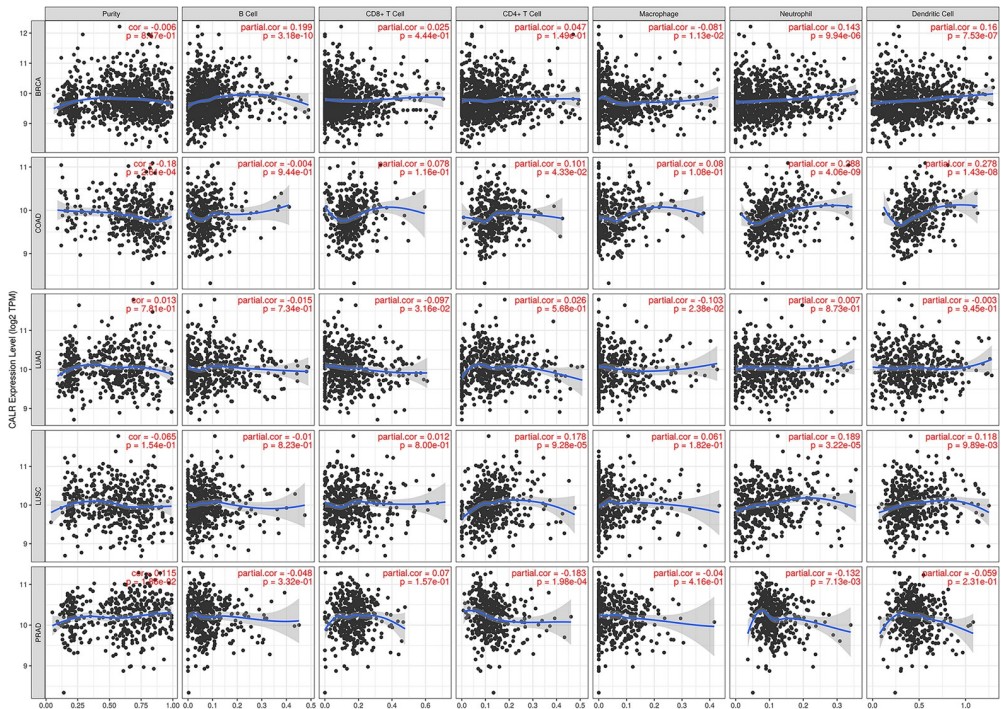

**Fig 8. Relationships between CALR and the immune cells infiltration in BRCA, COAD, LUSC, LUAD, and PRAD.** P value<0.05 is considered statistically significant.

play an anti-tumor role in colon and ovarian cancer by up-regulating B2M and CALR [46]. GANAB and CANX belong to endoplasmic reticulum protein chaperones, which cooperate with CALR to regulate endoplasmic reticulum stress in osteoarthritis and asthma, respectively [47, 48]. In triple-negative breast cancer patients, down-regulation of CALR and TAPBP tend to indicate a poor prognosis [49].Previous studies have shown that the carcinogenic mechanism of CALR is related to the exposure of malignant primordial cells to CALR, HSP70 and HSP90 on the plasma membrane [28]. HSP70 and HSP90 are important molecular chaperones to regulate oncoprotein stability and promote tumorigenesis [50]. In our study, HSP90 and CALR were detected to have an interactive relationship. The specific mechanism needs further experimental verification.

CALR plays a key role in anti-tumor immunity. Aging neutrophils and surviving cancer cells are susceptible to be labeled by CALR which is secreted from macrophages, thus releasing the "eat me" signal [51]. Through antigen-presenting cells (APC), DC "sense" immunogenicity and mediate phagocytosis. At the same time, DC provides sufficient co-stimulation to T cells to stimulate the production of tumor-specific CD8 [+] T cells [52]. It may explain why drugs that trigger CALR exposure can activate the immune system when combined with conventional chemotherapy, thus promoting cancer ICD [11], as reported anthracyclines and other apoptosis promoting drugs were used to treat colon cancer [12]. In our study, using 22 databases, we studied 39 common tumors. The results showed that CALR was associated with the expression of one or more immune cells in 35 tumors. These results demonstrate that CALR has great value as a potential target for immunotherapy.

Inevitably, there are still some limitations in our research. Our research mainly focused on the bioinformatics analysis of CALR expression and potential molecular mechanism in tumors, and we only verified the CALR expression in breast cancer cell lines in vitro. The

results of bioinformatics analysis need further experiments verification and explore the specific mechanism in vivo and in vitro.

## Conclusion

In conclusion, we comprehensively analyzed CALR from the perspective of public database and cell lines, and summarized the gene expression, clinical prognosis and molecular mechanism of CALR in different tumors. Our research suggests that CALR can be used as a biomarker to predict tumor prognosis and a potential target for tumor molecules and immunotherapy in specific tumors, which has the value for further deepen research.

## Supporting information

**S1 Fig. Pie chart showing the base-pair mutations of CALR in human cancers based on the COSMIC database.**
(TIF)

**S2 Fig.** (a) Mutation diagram of CALR across protein domains in different cancer types. (b) Mutation level of CALR in the TCGA database.
(TIF)

**S3 Fig. The correlation between CALR and the immune cell infiltration in 39 cancers.**
(TIF)

**S1 Checklist.**
(DOCX)

**S1 Table. Correlation analysis of related genes and CALR in tumors based on the Regulome Explorer.**
(XLSX)

## Author Contributions

**Conceptualization:** Yijun Li, Jianjun He.

**Data curation:** Yijun Li, Xiaoxu Liu, Heyan Chen, Peiling Xie, Rulan Ma.

**Formal analysis:** Yijun Li, Xiaoxu Liu, Heyan Chen, Peiling Xie.

**Funding acquisition:** Huimin Zhang.

**Investigation:** Yijun Li, Xiaoxu Liu, Rulan Ma.

**Methodology:** Heyan Chen, Peiling Xie, Rulan Ma.

**Project administration:** Rulan Ma.

**Resources:** Yijun Li, Heyan Chen.

**Supervision:** Peiling Xie, Rulan Ma, Jianjun He, Huimin Zhang.

**Validation:** Yijun Li, Xiaoxu Liu, Heyan Chen, Peiling Xie, Jianjun He, Huimin Zhang.

**Visualization:** Heyan Chen.

**Writing – original draft:** Yijun Li, Huimin Zhang.

**Writing – review & editing:** Jianjun He, Huimin Zhang.

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
