## [Decision Letter · Decision Letter 0]

28 Sep 2021

PONE-D-21-19634

Bioinformatics analysis for the role of CALR in human cancers 

PLOS ONE

Dear Dr. Zhang,

Thank you for submitting your manuscript to PLOS ONE. After careful consideration, we feel that it has merit but does not fully meet PLOS ONE’s publication criteria as it currently stands. Therefore, we invite you to submit a revised version of the manuscript that addresses the points raised during the review process.

We look forward to receiving your revised manuscript.

Kind regards,

Bing He

Academic Editor

PLOS ONE

2. ""PLOS requires an ORCID iD for the corresponding author in Editorial Manager on papers submitted after December 6th, 2016. Please ensure that you have an ORCID iD and that it is validated in Editorial Manager. To do this, go to ‘Update my Information’ (in the upper left-hand corner of the main menu), and click on the Fetch/Validate link next to the ORCID field. This will take you to the ORCID site and allow you to create a new iD or authenticate a pre-existing iD in Editorial Manager. Please see the following video for instructions on linking an ORCID iD to your Editorial Manager account: https://www.youtube.com/watch?v=_xcclfuvtxQ

Additional Editor Comments (if provided):

Reviewers' comments:

Reviewer's Responses to Questions

**Comments to the Author**

1. Is the manuscript technically sound, and do the data support the conclusions?

Reviewer #1: Yes

Reviewer #2: Partly

2. Has the statistical analysis been performed appropriately and rigorously? 

Reviewer #1: Yes

Reviewer #2: Yes

3. Have the authors made all data underlying the findings in their manuscript fully available?

Reviewer #1: Yes

Reviewer #2: Yes

4. Is the manuscript presented in an intelligible fashion and written in standard English?

Reviewer #1: No

Reviewer #2: No

5. Review Comments to the Author

Reviewer #1: This manuscript by Li et al, comprehensively analyzed the role of CALR in cancers using GeneCards, UALCAN, GEPIA, Kaplan-Meier Plotter, COSMIC, Regulome Explorer, String,GeneMANIA and TIMER databases. The authors assess the possibility of CALR as a potential therapeutic target and survival biomarker. They studied the CALR expression in normal human tissues, various tumors and tumor stages. They found that mutations of CALR are widely present in tumors. CALR interacted with different genes and co-expressed with various proteins. In tumors, a variety of immune cells are closely related to CALR.

Here are my comments:

1. Please check for grammatical errors in the manuscript.

2. For the Kaplan-Meier Plotter, the link does not work.

3. In the results for figures 2,3,4 and 5, authors should conclude each section with why are there differences in CALR in different types of cancer?

4. The interpretation is not clear for figure 6. Please fix this.

5. Please cite the following articles:

a) Nitika, Blackman J.S., Knighton L.E., Takakuwa J.E., Calderwood S.K, Truman A.W. Chemogenomic screening identifies the Hsp70 co-chaperone DNAJA1 as a hub for anticancer drug resistance. Sci Rep 10, 13831 (2020).

b) itka Fucikova, Iva Truxova, Michal Hensler, Etienne Becht, Lenka Kasikova, Irena Moserova, Sarka Vosahlikova, Jana Klouckova, Sarah E. Church, Isabelle Cremer, Oliver Kepp, Guido Kroemer, Lorenzo Galluzzi, Cyril Salek, Radek Spisek; Calreticulin exposure by malignant blasts correlates with robust anticancer immunity and improved clinical outcome in AML patients. Blood 2016; 128 (26): 3113–3124.

Reviewer #2: This study utilizes multiple bioinformatics methods, however it lacks logical connections between analyses. Also the conclusion needs to be experimental validated. Comments are the following:

1.Line 146 Page 7, “the tissues with upregulated CALR mRNA in all three databases”. The “upregulated” is confusing, cause it is not clear what the author are comparing and what is set as control here.

2.Line 155 Page 8, The list of tumors seems redundant. It is more appropriate to show it in the table. In addition, the abbreviation for the tumor name is repeated in the figure legend (Figure 2 & 3).

3.Line 213 Page 10, it makes no sense to enumerate the percentage of each mutation in the different tumors and is lengthy. What is the conclusion for this part? And please use table for data presentation.

4.Line 213 Page 10, “CALR can be detected to be related to other genes”. The statement is ambiguous. What is the relationship and What are the other genes?

5.Line 252 Page 12, PPI and co-expressed proteins are two independent aspects of protein studies. It is unreasonable to summarize them together.

6.In Figure 8, what are the criteria for determining whether CALR expression is related to immune cell infiltration? Moreover, does the CALR gene expression here refer to the tumor cells or the immune cells?

7.The writing of this manuscript needs to be improved. Errors in grammar and typing need to be corrected. Besides, a colloquial expression should be avoided and statements is more accurate in written scientific language.

8.The resolution of the figures needs to be improved.

9.Overall, the data in this study was numerous, but did not reach a clear conclusion in the end. I would suggest that CALR gene should be studied in one specific tumor and validated experimentally.

6. PLOS authors have the option to publish the peer review history of their article (what does this mean?). If published, this will include your full peer review and any attached files.

Reviewer #1: No

Reviewer #2: No

---

## [Author Response · Author response to Decision Letter 0]

23 Oct 2021

Response to Reviewers’ Comments

We thank the editor and reviewers for their very helpful comments. Our revisions in response to each of the reviewers’ comments have greatly improved the quality of the manuscript. All the revisions made in response to comments by the reviewers are highlighted in the revised manuscript. The responses to the reviewer’s comments are listed as following:

Reviewers’ Comments:

Reviewer #1

Comment:

This manuscript by Li et al, comprehensively analyzed the role of CALR in cancers using GeneCards, UALCAN, GEPIA, Kaplan-Meier Plotter, COSMIC, Regulome Explorer, String,GeneMANIA and TIMER databases. The authors assess the possibility of CALR as a potential therapeutic target and survival biomarker. They studied the CALR expression in normal human tissues, various tumors and tumor stages. They found that mutations of CALR are widely present in tumors. CALR interacted with different genes and co-expressed with various proteins. In tumors, a variety of immune cells are closely related to CALR.

Response:

We thank the reviewer for the very helpful comments. Our revisions in response to each of the comments (see below) have greatly improved the quality of the manuscript.

Comment:

1. Please check for grammatical errors in the manuscript.

Response: 

Thanks for your valuable suggestion. We are very sorry for our negligence. We have invited the native English speaker to further revise our manuscript. I hope our revised manuscript will relieve you of your concerns and meet with approval.

Comment:

2. For the Kaplan-Meier Plotter, the link does not work.

Response: 

Thank you for pointing this error out. We are very sorry for the incorrect writing. We have revised the link of the website in Page5, Line99.

Comment:

3. In the results for figures 2,3,4 and 5, authors should conclude each section with why are there differences in CALR in different types of cancer?

Response: 

Thanks for your valuable suggestion. We have added the explanation and summary for each part. Please find our revised version in the “Results” (Page8, Line164-168, Page9, Line189-192, Page10, Line202-204, Page10, Line216-217).

Comment:

4. The interpretation is not clear for figure 6. Please fix this.

Response: 

Thanks for your valuable suggestion. We have rewritten the “Genome-wide association of CALR in cancers” part of Results. Please find our revised version in the Page11, Line223-231.

Comment:

5. Please cite the following articles:

a) Nitika, Blackman J.S., Knighton L.E., Takakuwa J.E., Calderwood S.K, Truman A.W. Chemogenomic screening identifies the Hsp70 co-chaperone DNAJA1 as a hub for anticancer drug resistance. Sci Rep 10, 13831 (2020).

b) itka Fucikova, Iva Truxova, Michal Hensler, Etienne Becht, Lenka Kasikova, Irena Moserova, Sarka Vosahlikova, Jana Klouckova, Sarah E. Church, Isabelle Cremer, Oliver Kepp, Guido Kroemer, Lorenzo Galluzzi, Cyril Salek, Radek Spisek; Calreticulin exposure by malignant blasts correlates with robust anticancer immunity and improved clinical outcome in AML patients. Blood 2016; 128 (26): 3113–3124.

Response: 

Thanks for your valuable suggestion. We have cited the above articles as reference28 and reference50.

Reviewer #2

Comment:

This study utilizes multiple bioinformatics methods, however it lacks logical connections between analyses. Also the conclusion needs to be experimental validated. 

Response:

We thank the reviewer for the very helpful comments. Our revisions in response to each of the comments (see below) have greatly improved the quality of the manuscript.

Comment:

1. Line 146 Page 7, “the tissues with upregulated CALR mRNA in all three databases”. The “upregulated” is confusing, cause it is not clear what the author are comparing and what is set as control here.

Response: 

Thank you for pointing that out. We are very sorry for our unclear description. We have revised the interpretation of Figure 1 in the “Results” section according to your suggestion. Please find our revision in Page8, Lines155-158.

Comment:

2. Line 155 Page 8, The list of tumors seems redundant. It is more appropriate to show it in the table. In addition, the abbreviation for the tumor name is repeated in the figure legend (Figure 2 & 3).

Response: 

Thanks for your valuable suggestion. We have added Table 1 for data presentation based on your suggestion. In addition, we added the “Abbreviation” part after the conclusion to summarize the abbreviations in the full text.

Comment:

3. Line 213 Page 10, it makes no sense to enumerate the percentage of each mutation in the different tumors and is lengthy. What is the conclusion for this part? And please use table for data presentation.

Response: 

Thanks for your valuable suggestion. Based on your suggestion, we have added Table 2 for data presentation. Meanwhile, we conclude the findings for “CALR mutations in cancers”. Please find our revised version in the Page10, Line216-217.

Comment:

4. Line 213 Page 10, “CALR can be detected to be related to other genes”. The statement is ambiguous. What is the relationship and What are the other genes?

Response: 

Thank you for your suggestion. We have rewritten the “Genome-wide association of CALR in cancers” part of the Results. We defined this relationship as a pair of genes with P value < - log10 of the correlation between DNA methylation, somatic copy number, somatic mutation and protein level according to Spearman’s correlation analysis and displaying the associated genes in supplementary Table 1 in the revised manuscript. Please find our revision in the Page11, Line223-231.

Comment:

5. Line 252 Page 12, PPI and co-expressed proteins are two independent aspects of protein studies. It is unreasonable to summarize them together.

Response: 

Thanks for your valuable suggestion. We have rewritten the “PPI network of CALR” part of the Results. Please find our revised version in the Page12, Line235-241.

Comment:

6. In Figure 8, what are the criteria for determining whether CALR expression is related to immune cell infiltration? Moreover, does the CALR gene expression here refer to the tumor cells or the immune cells?

Response: 

Thank you for your suggestion. We are very sorry for the unclear description. CALR gene expression here refers to the tumor cells. And the correlation between CALR expression and immune cell infiltration was determined according to the Spearman’s test. The p value of <0.05 was considered statistically significant. We have added the detailed explanation in the Page7, Line133-137. 

Comment:

7. The writing of this manuscript needs to be improved. Errors in grammar and typing need to be corrected. Besides, a colloquial expression should be avoided and statements is more accurate in written scientific language.

Response: 

Thanks for your valuable suggestion. We are very sorry for our negligence. We have invited the native English speaker to further revise our manuscript. I hope our revised manuscript will relieve you of your concerns and meet with approval.

Comment:

8. The resolution of the figures needs to be improved.

Response: 

Thank you for your valuable suggestion. According to the publication requirements of the magazine, the maximum resolution of the figure is 600dpi. We have improved the resolution of all figures to 600dpi and had the clearer display as much as possible.

Comment:

9. Overall, the data in this study was numerous, but did not reach a clear conclusion in the end. I would suggest that CALR gene should be studied in one specific tumor and validated experimentally.

Response: 

Thanks for your valuable suggestion and we fully understand your concerns. In terms of research methods, firstly we introduced the molecular characteristics and differential expression of CALR in tumors and normal tissues. The survival analysis was performed between the expression level of CALR and clinical prognosis. The above studies were used to confirm the role of CALR in the occurrence of a variety of tumors. Subsequently, we performed mutation, Genome-wide association and PPI analysis to explore the possible molecular mechanism of CALR carcinogenesis. Finally, we analyzed the relationship between CALR and immune cells in different tumor cells to explore the CALR related tumor immune mechanism and the possibility of CALR as a target of immunotherapy. We've added our summary of research methods at the end of the “Materials and methods” part (Page7, Line140-146).

In addition, we have added the description at the beginning and the end of each section of the “Results” part to summarize the research purpose and results of each section, to improve logical connections between analyses. Please find our revision in the revised manuscript.

Regarding experimental validation, we have added in vitro data of CALR expression in breast cancer cell lines in the second part of “The expression of CALR in cancers” Page8-9, Line169-177), since breast cancer is the most common cancer worldwide. With regard to the lack of validation studies in more tumor cell lines, we added the explanation in the “limitations” part of the discussion (Page15, Line309-312).

In conclusion, in this study, we comprehensively analyzed CALR from the perspective of public database and clinical tumor samples, and summarized the gene expression, clinical prognosis and molecular mechanism of CALR in different tumors. Our research suggests that CALR can be used as a biomarker to predict tumor prognosis and a potential target for tumor molecules and immunotherapy, which has the value of further research. We have rewritten the “Conclusion” part in the Page15, Line315-319.

---

## [Decision Letter · Decision Letter 1]

29 Nov 2021

Bioinformatics analysis for the role of CALR in human cancers

PONE-D-21-19634R1

Dear Dr. Zhang,

We’re pleased to inform you that your manuscript has been judged scientifically suitable for publication and will be formally accepted for publication once it meets all outstanding technical requirements.

Kind regards,

Bing He

Academic Editor

PLOS ONE

Additional Editor Comments (optional):

Reviewers' comments:

Reviewer's Responses to Questions

**Comments to the Author**

1. If the authors have adequately addressed your comments raised in a previous round of review and you feel that this manuscript is now acceptable for publication, you may indicate that here to bypass the “Comments to the Author” section, enter your conflict of interest statement in the “Confidential to Editor” section, and submit your "Accept" recommendation.

Reviewer #1: All comments have been addressed

Reviewer #2: All comments have been addressed

2. Is the manuscript technically sound, and do the data support the conclusions?

Reviewer #1: Yes

Reviewer #2: Partly

3. Has the statistical analysis been performed appropriately and rigorously? 

Reviewer #1: Yes

Reviewer #2: Yes

4. Have the authors made all data underlying the findings in their manuscript fully available?

Reviewer #1: Yes

Reviewer #2: Yes

5. Is the manuscript presented in an intelligible fashion and written in standard English?

Reviewer #1: Yes

Reviewer #2: Yes

6. Review Comments to the Author

Reviewer #1: The authors have addressed my comments and I recommend the manuscript for publication in this revised form.

Reviewer #2: (No Response)

7. PLOS authors have the option to publish the peer review history of their article (what does this mean?). If published, this will include your full peer review and any attached files.

Reviewer #1: No

Reviewer #2: No

---

## [Editor Report · Acceptance letter]

6 Dec 2021

PONE-D-21-19634R1 

Bioinformatics analysis for the role of CALR in human cancers 

Dear Dr. Zhang:

I'm pleased to inform you that your manuscript has been deemed suitable for publication in PLOS ONE. Congratulations! Your manuscript is now with our production department. 

Kind regards, 

on behalf of

Dr. Bing He 

Academic Editor

PLOS ONE